# Recent Development of Prodrugs of Gemcitabine

**DOI:** 10.3390/genes13030466

**Published:** 2022-03-05

**Authors:** Bhoomika Pandit, Maksim Royzen

**Affiliations:** Department of Chemistry, University at Albany, 1400 Washington Ave., Albany, NY 12222, USA; bpandit@albany.edu

**Keywords:** gemcitabine, prodrug, chemotherapy, drug delivery, anticancer agent

## Abstract

Gemcitabine is a nucleoside analog that has been used widely as an anticancer drug for the treatment of a variety of conditions, including ovarian, bladder, non-small-cell lung, pancreatic, and breast cancer. However, enzymatic deamination, fast systemic clearance, and the emergence of chemoresistance have limited its efficacy. Different prodrug strategies have been explored in recent years, seeking to obtain better pharmacokinetic properties, efficacy, and safety. Different drug delivery strategies have also been employed, seeking to transform gemcitabine into a targeted medicine. This review will provide an overview of the recent developments in gemcitabine prodrugs and their effectiveness in treating cancerous tumors.

## 1. Introduction

Gemcitabine (GCB) is a nucleoside analog that has been widely used as an antimetabolite antineoplastic agent. GCB has been approved by the US FDA since 1996, and has been sold under the brand name Gemzar. It is administered alone or in combination with other anticancer drugs to treat a variety of conditions, including ovarian, bladder, non-small-cell lung, pancreatic, and breast cancer [1,2]. At the cellular level, GCB is internalized via nucleic acid transporters. It is subsequently phosphorylated by dioxycytidine kinase (DCK). The stepwise phosphorylation leads to the formation of GCB-triphosphate, which is incorporated into cellular DNA, thereby inhibiting nuclear replication.

The clinical efficacy of GCB is affected by a number of well-documented limitations. Enzymatic deamination by cytidine deaminase (CDA) can lead to the formation of inactive 2′-deoxy-2′,2′-difluorouridine (dFdU) [3]. Cancer cells can acquire chemoresistance by the decreased expression of nucleoside transporters, especially the human equilibrative nucleoside transporter (hENT) [4]. Poor tumor targeting can lead to undesirable adverse effects. To address these challenges, multiple attempts have been made to develop prodrugs of GCB that would be stable during enzymatic inactivation, have improved pharmacokinetic properties, and could be employed for targeted cancer therapy. This review will describe the chemical strategies that have been reported in recent years.

## 2. Prodrugs of Gemcitabine

### 2.1. Hydrophobic Prodrugs with Different Mechanisms of Cellular Uptake

Huixin Qi et al. reported a small library of prodrug compounds, shown in Figure 1, that contain hydrophobic monophosphate ester modifications at the 5′-position of GCB [5]. In the prodrug form, these compounds were expected to be cell permeable, aided by their long hydrophobic tails. Unexpectedly, the reported compounds were found to enter the cells via a different mechanism than GCB. It has previously been determined that GCB utilizes nucleic acid transporters, such as hENT, to achieve cell permeability. The authors showed that inhibiting hENT using dipyridamole did not significantly impact cellular uptake of the prodrugs, suggesting that they likely enter the cell by passive diffusion through the cellular membrane. This is potentially advantageous, as downregulation of hENT can cause cellular chemoresistance to GCB [6]. Inside the cell, the prodrugs were expected to be enzymatically converted to the 5′-monophosphate of GCB and, ultimately, incorporated into the cellular DNA. The reported compounds were evaluated in vitro against a variety of cancer cells using an MTT assay. Prodrug **3** showed the best potency, which was comparable to the cytotoxicity of GCB, especially in A549, DU145, and PC-3 cell lines.

The hydrophobic tail of prodrug **3** reduced its solubility in aqueous media. To achieve therapeutically meaningful doses, the prodrug had to be formulated in a saline solution containing 1% CMC-Na, with 0.5% Tween 80. Pharmacokinetic parameters were evaluated in nude mice after oral administration of prodrug **3** and GCB. The prodrug dose of 20 mg/kg (9.4 mg/kg GCB equiv.) delivered similar systemic exposure as compared to the parent drug. The in vivo efficacy was investigated using a H460 tumor xenograft model (non-small-cell lung cancer). Four oral prodrug doses of 40 mg/kg (18.7 mg/kg GCB equiv.), given once every 3 days, resulted in 65.2% tumor suppression. By comparison, four IP GCB doses of 80 mg/kg, given once every 3 days, resulted in 61.1% tumor suppression. Thus, lower doses of the prodrug were able to achieve comparable efficacy to GCB.

### 2.2. Prodrugs That Respond to Enzymatic Activation

Xinming Li et al. designed a prodrug of GCB, termed **S-Gem** (Figure 2A), whose enzymatic activation can be triggered by thioredoxin reductase (TrxR) [7]. Malignant cancer cells have abnormally high levels of TrxR for maintaining their tumor phenotypes [8]; thus, it was hypothesized that prodrug activation would be selectively achieved inside the cancer cells. The key residue of TrxR is selenocysteine (Sec). It is an analogue of cysteine in which the sulfur atom is replaced by selenium. A Sec residue has a significantly lower pKa value than that of the thiol group in Cys (5.8 for Sec and 8.3 for Cys), which renders the selenol group in Sec to be predominantly present as selenolate. The authors claim that **S-Gem** activation is drastically lower if the Sec residue of TrxR is replaced by Cys. The proposed mechanism of activation of S-Gem by TrxR is illustrated in Figure 2B. The enzyme reduces the disulfide bond of 1,2-dithiolane, which leads to intramolecular cyclization and subsequent activation of GCB.

To understand the mechanism of prodrug activation, the authors carried out HPLC studies. Incubation of **S-Gem** in a buffered media containing TrxR resulted in about 80% release of activated GCB after 4 h. The authors synthesized a negative control analog of the prodrug, in which both the sulfur atoms were replaced by carbons (**C-Gem**). Incubation of **C-Gem** with the artificial reducing agent TCEP (tris(2-carboxyethyl)phosphine) produced a small amount of activated GCB. Meanwhile, the incubation of **S-Gem** with TCEP resulted in a nearly quantitative amount of GCB. Interestingly, the authors also reported that reduced TrxR is capable of activating **S-Gem** to produce GCB. However, this unexpected finding was not investigated any further. Subsequently, prodrug activation was studied in HeLa cells, which express TrxR. Prodrug activation has been observed in HeLa cell lysates. However, inhibiting the enzyme with auranofin considerably decreased prodrug activation.

The cytotoxicity studies of **S-Gem** and **C-Gem** (negative control) were carried out in the following three cell lines: SMMC-7721 cells (human hepatocellular carcinoma), A549 cells (adenocarcinomic human alveolar basal epithelial cells), and HeLa cells (human cervical cancer cells). MTT studies determined that the IC_50_ values were 1.4 μM for SMMC-7721, 0.6 μM for A549, and 2.2 μM for HeLa cells. Meanwhile, **C-Gem** was virtually non-toxic. To confirm that the cytotoxicity **of S-Gem** is dependent on TrxR, the authors performed MTT studies using HeLa cells in which the enzyme was knocked down. Consistent with the proposed hypothesis, the knock down of TrxR resulted in, roughly, a 10-fold increase in the observed IC_50_ value.

The authors did not evaluate their prodrug in vivo. It would be very interesting to investigate the prodrug’s in vivo behavior. A number of critical elements would be at play. Firstly, the solubility of the prodrug is likely to be lower than that of GCB, which is administered as an HCl salt. Secondly, the prodrug will likely be more lipophilic, which will affect its pharmacokinetic properties. Lastly, the stability of the carbamate group in blood plasma will be highly important.

### 2.3. Prodrugs That Are Activated by ROS

Matsushita et al. developed a prodrug of GCB, termed **A-Gem** (Figure 3), which releases the active anticancer drug in the presence of H_2_O_2_ [9]. Oxidative stress is a common feature of many cancers, resulting in elevated levels of reactive oxygen species (ROS) compared to normal cells [10,11,12,13]. H_2_O_2_ is an example of ROS, which is known for its cell permeability and stability [14,15,16,17]. The reported prodrug design is based on the following well-established chemistry: reduction of alkyl and arylboronic acids by H_2_O_2_ [18]. This reaction is known to be bio-orthogonal and biocompatible. The prodrug **A-Gem** follows the same reaction mechanism, as it contains a boronate-ester carbamate group at the N4-position of the nucleobase (Figure 3). The authors also reported a structurally similar negative control compound, **6**, which does not have the boronate-ester linkage (Figure 3).

The authors evaluated the mechanism behind the uncaging of **A-Gem** by H_2_O_2_ using HPLC analysis. When **A-Gem** was treated with an equimolar amount of H_2_O_2_, under simulated physiological conditions, GCB was observed within 20 min, with 95% conversion. On the other hand, **6** was not converted into GCB under the same treatment. Interestingly, the activation of **A-Gem** was shown to be highly selective for H_2_O_2_ compared to other ROS present in the human body, such as tert-butylhydroperoxide (TBHP), hypochlorite (ClO^−^), hydroxyl radical (OH^−^), tert-butoxy radical (tBuO^−^), nitric oxide (NO), and superoxide (O_2_^−^).

The investigation of the cytotoxicity of GCB, **A-Gem,** and **6** was carried out in human pancreatic cancer cell lines (PSN1 and BxPC3) and a normal pancreatic epithelial cell line (NPEC), using an MTT assay. As anticipated, the cytotoxicity of **6** was considerably lower that of **A-Gem** in the cancer cells. On the other hand, their cytotoxicity in NPEC cells, which have lower levels of H_2_O_2_, was essentially the same. The addition of exogenous H_2_O_2_ resulted in higher cytotoxicity of **A-Gem**, thus indicating that the tested cancer cells do not have a sufficiently high endogenous concentration of H_2_O_2_ to efficiently activate all of the dosed prodrug.

The in vivo evaluation of therapeutic efficacy was carried out in the following three cohorts of mice: GCB, **A-Gem**, and the vehicle. The **A-Gem** group received a two-times higher dose (100 mg/kg) than the GCB group (50 mg/kg). The mice were dosed four times with GCB, **A-Gem**, and the vehicle on days 14, 17, 20, and 23. The prodrug had statistically analogous efficacy to GCB. The average tumor suppression was essentially the same in the **A-Gem** and GCB cohorts. On the other hand, myelosuppression, the most common side effect of GCB therapy, was lower in the prodrug cohort. LC-MS analysis of the tumor tissue showed similar levels of the activated drug in the **A-Gem** and GCB cohorts. On the other hand, LC-MS analysis of the bone marrow indicated lower amounts of the activated drug in the prodrug cohort. These findings support the hypothesis that **A-Gem** behaves as a tumor-selective anticancer agent.

### 2.4. Prodrugs That Attempt to Counter Multidrug Resistance

Hong et al. reported prodrugs of GCB that contain various amino acids conjugated to the N4-position of the nucleobase of GCB (Figure 4) [19]. This strategy aims to address one of the mechanisms of multidrug resistance (MDR) triggered by the overexpression of various efflux transporters by cancer cells [20,21,22]. The strategy takes advantage of the concomitant overexpression of amino acid uptake transporters by cancer cells undergoing neoplastic transformation [23]. The authors, in fact, confirmed that the levels of the amino acid transporter LAT-1 were elevated in the pancreatic cancer cell lines BxPC-3 and MIAPaCa-2. The LAT-1 transporter is persistently found in patients with pancreatic cancer, and in transplanted Colo357 cells (pancreatic cancer cell line) [24,25]. The authors hypothesized that using an amino acid-conjugated prodrug of GCB might be an effective way to improve GCB uptake via transporters, such as LAT-1. A threonine derivative of GCB (**Gem-Thr**) would be transported inside the cancer cells be activated by LAT-1 via amide bond cleavage.

The authors investigated the viability of various cancer cells treated with 1 mM prodrugs for 48 h. **Gem-Thr** showed a similar profile to GCB, being the most cytotoxic against BxPC-3 and B16 cell lines. On the other hand, it was mostly non-toxic against MIAPaCa-2, A549, and MDA-231 cells. The authors performed a TUNEL assay, which suggested that apoptosis was induced by GCB and **Gem-Thr** treatments.

The stability of **Gem-Thr** was investigated in PBS, rat blood plasma, and liver microsomes using LC-MS. After incubation for 8 h at 37 °C, the prodrug was found to be remarkably metabolically stable in PBS and rat liver microsomes, even in the presence of elevated cytochrome P450. In rat blood plasma, 20% of **Gem-Thr** was converted to GCB.

The pharmacokinetics parameters of GCB and **Gem-Thr** were investigated in rats by IV administration of a 4 mg/kg dose. The AUC and the measure of drug elimination from the body (CL) were found to be 948.38 ± 52.04 μg·min/mL and 4.23 ± 0.23 mL/min/kg, respectively. The volume of distribution at steady state (Vss) and the average time for a drug molecule to reside in the body (MRT), for GCB, were 2483.64 ± 867.19 mL/kg and 582.06 ± 177.90 min, respectively. After administration of **Gem-Thr** at 4 mg/kg IV, the conversion of **Gem-Thr** to GCB was found to be likely due to amide bond cleavage. More importantly, **Gem-Thr** increased systemic exposure (i.e., as defined by the AUC of GCB) by 1.83-fold versus free GCB. This was attributed to the significantly lower total CL value (0.60 vs. 4.23 mL/min/kg). These findings suggest that the amide prodrug approach improves metabolic stability in vivo.

### 2.5. Macromolecular Prodrugs of Gemcitabine

Zhang et al. reported macromolecular constructs containing albumin-conjugated prodrugs of GCB, termed **Albumin-1a** and **Albumin-1b**. The reported compounds contain albumin conjugated to 5′-OH of GCB via a carbonate linker [26]. Analogous prodrugs with albumin conjugated to the N4-position of the cytosine nucleobase via a carbamate linker were also described. GCB has a relatively short half-life in vivo. Within 9 min in the human body, it undergoes deamination of the cytosine nucleobase by cytidine deaminase (CDA), which results in the formation of the inactive form of GCB, 2′-deoxy-2′,2′-difluorouridine (dFdU) [27]. To overcome this limitation and enhance its efficacy, human serum albumin (HSA) was used as a macromolecular drug carrier. HSA remains in the blood plasma for days after administration. The reported prodrugs can be activated by the following two different mechanisms, as illustrated in Figure 5: (1) hydrolysis of the labile carbonate linkage at 5′-OH of the sugar moiety; (2) reduction of the disulfide linkage by GSH, followed by intramolecular cyclization.

The studies of albumin binding to prodrugs were carried out with bovine serum albumin (BSA). All of the reported prodrugs attached to BSA very fast. Within 1 min of incubation, over 90% of the prodrugs had coupled to BSA. Complete binding was achieved after 30 min of incubation. There was no significant difference between the circular dichroism (CD) data for BSA before and after conjugation, suggesting that the introduction of the prodrug to BSA did not cause significant changes to its secondary structure.

The authors used dithiothreitol (DTT) to study the reductive mechanism of prodrug activation. **Albumin-1b** was treated with 1 μM DTT for 8 h in PBS (pH = 7.4). This treatment resulted in 33% release of the active form of GCB. A high concentration of DTT (1 mM) drastically improved the release to 100%. The stability of the prodrugs against deamination by CDA was investigated in rat blood plasma. The prodrugs and GCB were incubated at 37 °C for 12 h at a concentration of 16 μg/mL. It was observed that **Albumin-1b** was mostly transformed to active GCB and formed three-times less dFdU than GCB alone. The other three prodrugs produced no more than 380 ng/mL of dFdU per hour. These results strongly suggest that the prodrugs are less prone to degradation by CDA than GCB.

Antitumor efficacy was investigated in mice bearing 4T1 xenografts. The prodrugs and GCB were administered in five injections of an equivalent dose of 8 mg/kg GCB. Unexpectedly, the prodrug **Albumin-1a**, which contains a carbonate linkage bond and an all-carbon linker lacking a disulfide bond, was found to be the most potent anticancer agent. Treatment with **Albumin-1b** resulted in a reduction in the tumor volume by two-thirds. The authors attributed this result to several factors, as follows: (1) improved pharmacokinetic properties; (2) decreased prodrug deactivation by CDA; (3) enhanced prodrug accumulation in the tumor; (4) improved cellular uptake that is less dependent on nucleic acid transporters; (5) in vivo prodrug activation via cleavage of the carbonate bond.

### 2.6. Prodrugs That Are Based on Bio-Orthogonal Chemistry

Weiss et al. reported prodrugs of GCB, **7** and **8**, which can be uncaged by bio-orthogonal palladium Pd^0^ chemistry [28]. Transition metal-assisted bio-orthogonal chemistry could be helpful for the controlled release of the drug, which will enhance its efficacy. The authors reported six compounds containing allyl (Alloc), propargyloxycarbonyl (Poc), and carboxybenzyl (Cbz) groups attached to GCB via either a carbamate or carbonate linker (Figure 6). The authors proposed that protecting the 5′-OH would block the formation of cytotoxic metabolites, while masking the N4-position of cytidine would prevent deactivation by CDA.

The authors carried out cytotoxicity studies using two human cancer cell lines, BxPC3 and Mia PaCa-2. Prodrugs **8 a**, **p**, and **b** were found to be less cytotoxic than GCB, indicating that the carbamate linker has a shielding effect. On the other hand, prodrugs **7 a**, **p, and b** exhibited similar cytotoxicity to the parent drug. These results indicated that the carbonate linker is cleaved inside the cells, leading to non-specific prodrug activation. Therefore, only compounds **8 a, p, and b** were used for further investigation.

The authors carried out HPLC analysis to investigate the release of active GCB from prodrugs **8 a**, **p**, and **b**. Pd^0^-functionalized resins were prepared by trapping Pd^0^ nanoparticles in an amino-functionalized polystyrene matrix. To monitor the effectiveness of Pd^0^ catalysis, all carbamates **8 a, p, and b** (100 μM) were incubated with Pd^0^ resins (1 mg/mL) dispersed in PBS (300 mOsm/kg, pH = 7.4) at 37 °C for 24 h. As per the analysis of retention times, compound **8 b** only generated negligible amounts of active GCB, while **8 a** and **8 p** produced notable amounts of active GCB. Compound **8 p** showed fast and robust unmasking of the carbamate group, with a half-life of less than 6 h. This was consistent with the previously reported palladium-driven catalysis of propargyl groups [29].

The Pd^0^-catalyzed bio-orthogonal in situ generation of active GCB from the prodrugs **8 a, p**, **and b** was further studied in BxPC-3 and Mia PaCa-2 cell lines. The cells were treated with the prodrugs and Pd^0^ resins, and the observed cytotoxicities were compared to GCB. As anticipated, prodrugs **8 a** and **8 p** showed the strongest effect, while prodrug **8 b** was not activated by Pd^0^. Among all the tested compounds, **8 p** had the highest potential to generate active GCB in the Pd^0-^catalyzed bio-orthogonal reaction. In conclusion, the authors emphasized that the bio-orthogonal reaction between the carbamate prodrugs and Pd^0^ has great potential as a targeted cancer therapy.

Our group, in collaboration with Shasqi, Inc., recently reported a prodrug of GCB, shown in Figure 7, whose activation is triggered by bio-orthogonal inverse electron demand Diels-Alder (IEDDA) chemistry between *trans*-cyclooctene (TCO) and tetrazine (Tz) [30]. The prodrug, termed **GCB-TCO-acid**, contains TCO attached to the N4-position of the nucleobase via a carbamate bond. In the caged form, the prodrug is shielded from deamination by CDA. A carboxylic acid moiety was installed on TCO to improve the prodrug’s aqueous solubility. Upon reaction with Tz, the prodrug is expected to be activated via the bond-cleaving bio-orthogonal chemistry mechanism described in Figure 7 [31].

We carried out an assessment of the prodrug’s in vitro properties to determine that it is stable in mouse blood plasma and has a solubility of 3 mg/mL in PBS. The MTT studies with MC38 cells (murine colon adenocarcinoma) showed that the prodrug was 8.7-times less cytotoxic than GCB (IC_50_ = 3 nM for GCB; IC_50_ = 26 nM for **GCB-TCO-acid**). We envisioned that **GCB-TCO-acid** could be used in conjunction with the CAPAC platform, as shown in Figure 8. The biomaterial-based Click Activated Protodrugs Against Cancer (CAPAC) platform consists of a tetrazine-modified sodium hyaluronate-based biopolymer, **SQL70**, injected near the tumor site, followed by systemic doses of **GCB-TCO-acid**. The prodrug is captured locally by the biopolymer through an IEDDA reaction, followed by conversion to active GCB at the tumor site.

We evaluated the antitumor effects of **GCB-TCO-acid** in combination with the reported Tz-functionalized biopolymer **SQL70** [32]. This study was carried out in immune-competent mice bearing MC38 xenografts. **GCB-TCO-acid** was administered IV at a dose of 55 mg/kg, once daily, for five consecutive days. The prodrug treatment correlated with the anti-tumor efficacy (*unpublished data*). However, because of inadequate inactivation, the administration of high doses of **GCB-TCO-acid** was associated with severe neurotoxicity. Further work will have to be conducted in the future to design a prodrug of GCB with a higher level of inactivation that can be tolerated more safely at high doses.

## 3. Conclusions

In conclusion, this review described a number of different prodrug strategies aimed at addressing the shortcomings of FDA-approved GCB-based chemotherapy, such as enzymatic deamination, fast systemic clearance, and chemoresistance by downregulation of cellular uptake. The described strategies included GCB modification with enzyme-labile groups, groups that are cleavable by tumor-specific environmental factors, such as ROS, as well as macromolecular approaches and drug delivery approaches. The modification of GCB at the 5′-OH position or N4-position of the cytosine nucleobase can lead to significant deactivation, as was shown by the in vitro MTT studies. The reported carbamate prodrugs tend to be more stable under simulated physiological conditions, while the reported carbonate prodrugs are less stable. However, they presented alternative in vivo activation strategies that led to promising in vivo data. Several of the described reports presented in vivo efficacy data showing that prodrugs of GCB have fewer side effects and can achieve greater anti-tumor efficacy. The reported data offer significant promise that more effective GCB-based anticancer therapies will be developed in the future.

## Figures and Tables

**Figure 1 genes-13-00466-f001:**
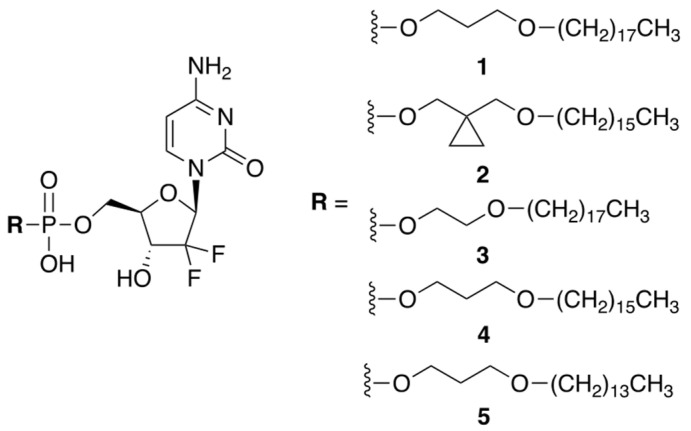
A library of prodrugs of GCB containing hydrophobic monophosphate ester modifications.

**Figure 2 genes-13-00466-f002:**
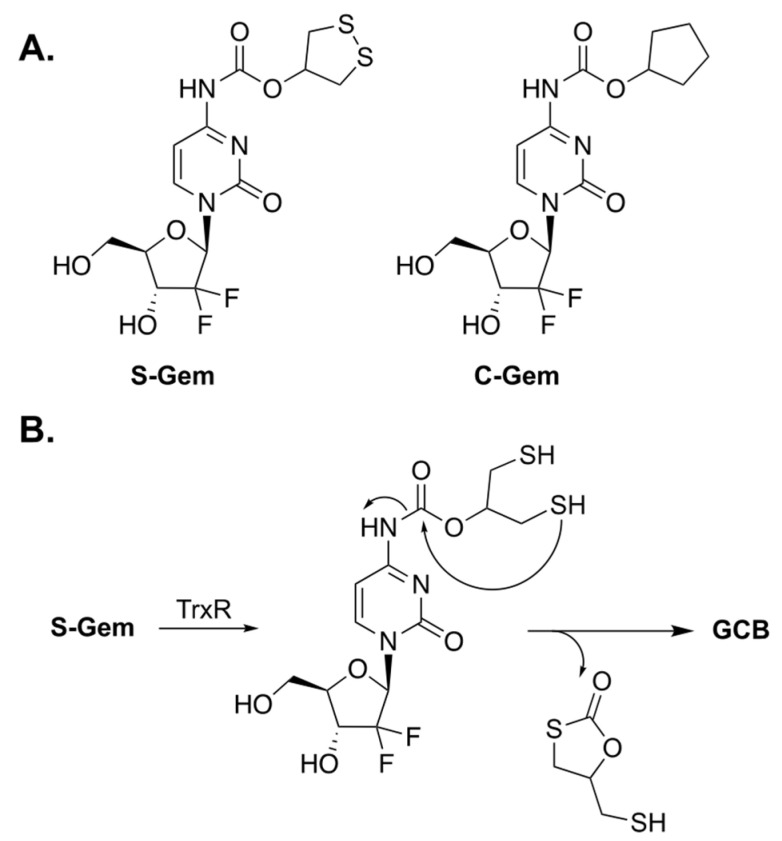
(**A**) Prodrugs of GCB, **S-Gem** and **C-Gem**. (**B**) Mechanism of enzymatic activation of **S-Gem**.

**Figure 3 genes-13-00466-f003:**
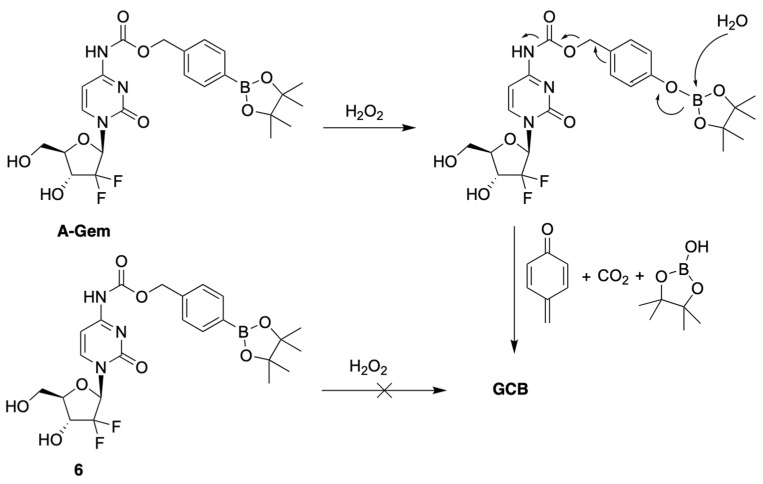
Mechanism of activation of **A-Gem** by hydrogen peroxide.

**Figure 4 genes-13-00466-f004:**
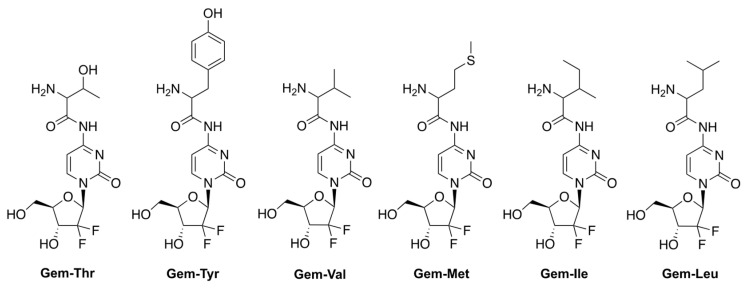
Prodrugs of GCB, containing various amino acids conjugated to the N4-position of the nucleobase.

**Figure 5 genes-13-00466-f005:**
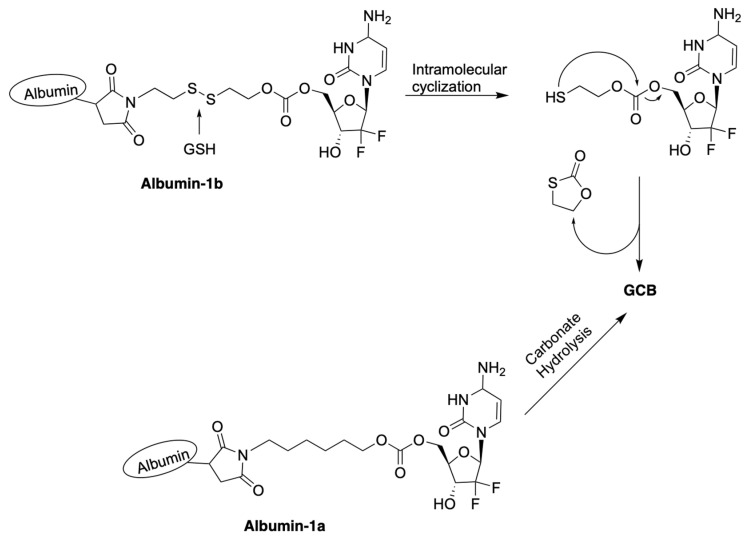
Proposed mechanisms of activation of prodrugs **Albumin-1a** and **Albumin-1b**.

**Figure 6 genes-13-00466-f006:**
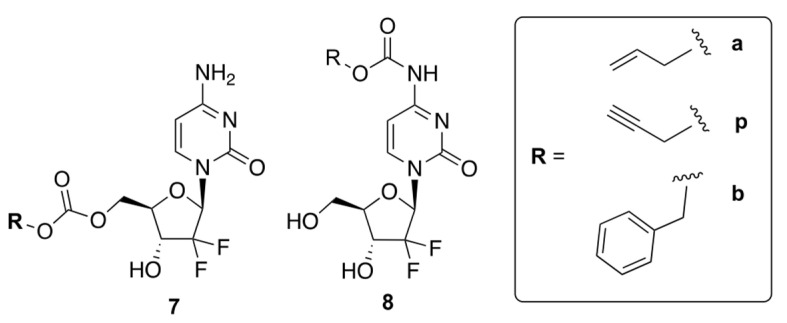
Prodrugs of GCB that can be uncaged by bio-orthogonal palladium Pd^0^ chemistry.

**Figure 7 genes-13-00466-f007:**
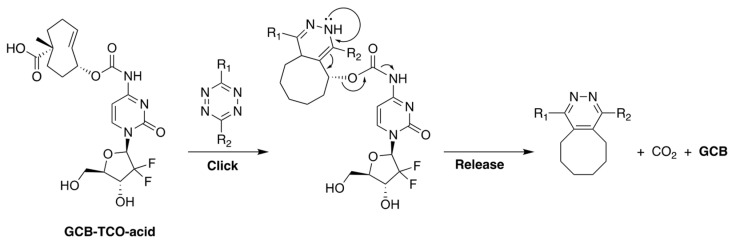
The structure of **GCB-TCO-acid** and the mechanism of its activation by Tz via bond-cleaving bio-orthogonal IEDDA chemistry.

**Figure 8 genes-13-00466-f008:**
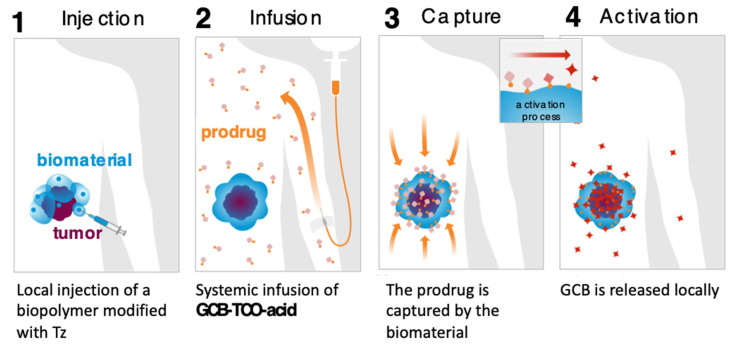
CAPAC platform for local activation of systemically administered prodrugs to treat cancer.

## Data Availability

Not applicable.

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
