# Peer review of "Recent Development of Prodrugs of Gemcitabine"

_genes, 2022, doi:10.3390/genes13030466_

Round 1

Reviewer 1 Report

This Review describes a number of different prodrug strategies aimed at addressing shortcomings of FDA-approved GCB-based chemotherapy, such as enzymatic deamination, fast systemic clearance and chemoresistance by downregulation of cellular uptake.

The authors report encouraging data that а more effective GCB-300-based anticancer therapy can be developed in the future.

 Comments: Please take into consideration the followings:

The Fig.1 (line 61) is of low quality, and should be improved.

The Fig.2 (line 89) is too large and should be reduced.

The manuscript provides a good amount of data and it is relevant to the journal.

Author Response

Following the reviewer's suggestion, Figure 1 was modified and Figure 2 was reduced in size.

Reviewer 2 Report

The author summarized the different prodrug strategies of the GCB-based chemotherapy. Not only do the described strategies benefit the GCB development as shown in this review, they surely will provide insights for other prodrugs development. The article is well organized and comprehensive.

One thing needed to be revised:

The GCB is a chiral compound, the author should draw the absolute structure of GCB in the Figs 1-6.

Author Response

All figures were revised to indicate the chirality of gemcitabine.